# Understanding Non-linearity in Graph Neural Networks from the Perspective of Bayesian Inference

**Rongzhe Wei[1], Haoteng Yin[1], Junteng Jia[2], Austin R. Benson[2], Pan Li[1]**
[1] Department of Computer Science, Purdue University
[2] Department of Computer Science, Cornell University
{wei397, yinht, panli}@purdue.edu, jj585@cornell.edu, arbenson@gmail.com

## Abstract

Graph neural networks (GNNs) have shown superiority in many prediction tasks over graphs due to their impressive capability of capturing nonlinear relations in graph-structured data. However, for node classification tasks, often, only marginal improvement of GNNs over their linear counterparts has been observed. Previous works provide very few understandings of this phenomenon. In this work, we resort to Bayesian learning to deeply investigate the functions of non-linearity in GNNs for node classification tasks. Given a graph generated from the statistical model CSBM, we observe that the max-a-posterior estimation of a node label given its own and neighbors' attributes consists of two types of non-linearity, a possibly non-linear transformation of node attributes and a ReLU-activated feature aggregation from neighbors. The latter surprisingly matches the type of non-linearity used in many GNN models. By further imposing a Gaussian assumption on node attributes, we prove that the superiority of those ReLU activations is only significant when the node attributes are far more informative than the graph structure, which nicely matches many previous empirical observations. A similar argument can be achieved when there is a distribution shift of node attributes between the training and testing datasets. Finally, we verify our theory on both synthetic and real-world networks. Our code is available at https://github.com/Graph-COM/Bayesian_inference_based_GNN.git.

## 1 Introduction

Learning on graphs (LoG) has been widely used in the applications with graph-structured data [1,2]. Node classification, as one of the most crucial tasks in LoG, asks to predict the labels of nodes in a graph, which has been used in many applications such as community detection [3–6], anomaly detection [7,8], biological pathway analysis [9,10] and so on.

Recently, graph neural networks (GNNs) have become the de-facto standard used in many LoG tasks due to their super empirical performance [11,12]. Researchers often attribute such success to non-linearity in GNNs which associates them with great expressive power [13,14]. GNNs can approximate a wide range of functions defined over graphs [15–17] and thus excel in predicting, e.g., the free energies of molecules [18], which are by nature non-linear solutions of some quantum-mechanical equations. However, for node classification tasks, many studies have shown that non-linearity to control the exchange of features among neighbors seems not that crucial. For example, many works use linear propagation of node attributes over graphs [19,20], and others recommend adding non-linearity while only to the transformation of initial node attributes [21–23]. Both cases achieve comparable or even better performance than other models with complex nonlinear propagation, such as using neighbor-attention mechanism [24]. Recently, even in the complicated heterophilic setting where nodes with same labels are not directly connected, linear propagation still achieves competitive performance [25,26], compared with the models with nonlinear and deep architectures [27,28].

36th Conference on Neural Information Processing Systems (NeurIPS 2022).

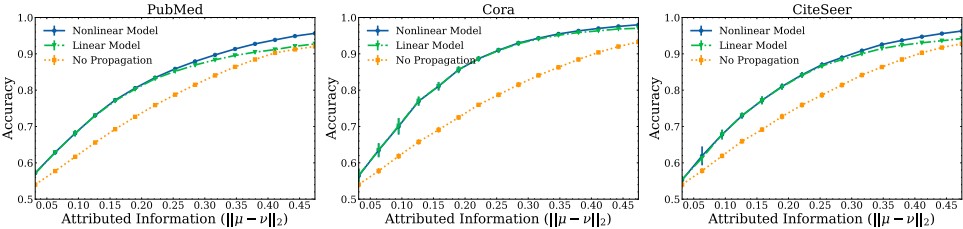

Figure 1: Averaged one-vs-all Classification Accuracy on Citation Networks of Nonlinear Models v.s. Linear Models. Node attributes in or out of the one class are generated from Gaussian distributions $\mathcal{N}(\mu, \frac{I}{m})$ and $\mathcal{N}(\nu, \frac{I}{m})$, $\mu, \nu \in \mathbb{R}^m$, respectively. The detailed settings are introduced in Sec. 5.3.

Although empirical studies on GNNs are extensive till now and many practical observations as above have been made, there have been very few works attempting to characterize GNNs in theory, especially to understand the effect of non-linearity by comparing with the linear counterparts for node classification tasks. The only work on this topic to the best of our knowledge still focuses on comparing the expressive power of the two methods to distinguish nodes with different local structures [29]. However, the achieved statement that non-linear propagation improves expressiveness may not necessarily reveal the above phenomenon that non-linear and linear methods have close empirical performance while with subtle difference. Moreover, more expressiveness is often at the cost of model generalization and thus may not necessarily yield more accurate prediction [30, 31].

In this work, we expect to give a more precise characterization of the values of non-linearity in GNNs from a statistical perspective, based on Bayesian inference specifically. We resort to contextual stochastic block models (CSBM) [32, 33]. We make a significant observation that given a graph generated by CSBM, the max-a-posterior (MAP) estimation of a node label given its own and neighbors' features surprisingly corresponds to a graph convolution layer with *ReLU* as the activation combined with an initial node-attribute transformation. Such a transformation of node attributes is generally nonlinear unless they are generated from the natural exponential family [34]. Since the MAP estimator is known to be Bayesian optimal [35], the above observation means that ReLU-based propagation has the potential to outperform linear propagation. To precisely characterize such benefit, we further assume that the node attributes are generated from a label-conditioned Gaussian model, and analyze and compare the node mis-classification errors of linear and nonlinear models. We have achieved the following conclusions (note that we only provide informal statements here and the formal statements are left in the theorems).

- When the node attributes are less informative compared to the structural information, non-linear propagation and linear propagation have almost the same mis-classification error (case I in Thm. 2).
- When the node attributes are more informative, non-linear propagation shows advantages. The mis-classification error of non-linear propagation can be significantly smaller than that of linear propagation with sufficiently informative node attributes (case II in Thm. 2).
- When there is a distribution shift of the node attributes between the training and testing datasets, non-linearity provides better transferability in the regime of informative node attributes (Thm. 3).

Given that practical node attributes are often not that informative, the advantages of non-linear propagation over linear propagation for node classification is limited albeit observable. Our analysis and conclusion apply to both homophilic and heterophilic settings, i.e., when nodes with same labels tend to be connected (homophily) or disconnected (heterophily), respectively [25, 27, 28, 36, 37].

Extensive evaluation on both synthetic and real datasets demonstrates our theory. Specifically, the node mis-classification errors of three citation networks with different levels of attributed information (Gaussian attributes) are shown in Fig. 1, which precisely matches the above conclusions.

## 1.1 More Related Works

GNNs have achieved great empirical success while theoretical understanding of GNNs, their non-linearity in particular, is still limited. There are many works studying the expressive power of GNNs [16, 38–48], while they often assume arbitrarily complex non-linearity with limited quantitative results. Only a few works provide characterizations on the needed width or depth of GNN layers [45–48]. More quantitative arguments on GNN analysis often depend on linear or Lipschitz continuous

assumptions to enable graph spectral analysis, such as feature oversmoothing [49, 50] and over-squashing [51, 52], the failure to process heterophilic graphs [25, 27, 53] and the limited spectral representation [54, 55]. Some works also study the generalization bounds [56–58] and the stability of GNNs [59–62]. However, the obtained results may not reveal a direct comparison between non-linearity and linearity of the model, and their analytic techniques avoid tackling the specific forms of non-linear activations by using a Lipschitz continuous bound which is too loose in our case.

Stochastic block models (SBM) and its contextual counterparts have been widely used to study the node classification problems [32, 33, 63–67], while these studies focus on the fundamental limits. Recently, (C)SBM and its large-graph limitation also have been used to study the transferrability and expressive power of GNN models [68–70] and GNNs on line graphs [5], while these works did not compare non-linear and linear propagation. CSBM has also been used to show the advantage of linear convolution over no convolution for node classification [71]. A very recent result shows that attention-based propagation [24] may be much worse than linear propagation given low-quality node attributes under CSBM [72]. Our results imply that ReLU is the de facto optimal non-linearity instead of attention and may at most marginally outperform the linear model when with low-quality node attributes. Some previous works also use Bayesian inference to inspire GNN architectures [73–80], while these works focus on empirical evaluation instead of theoretical analysis.

## 2 Preliminaries

In this section, we introduce preliminaries and notations for our later discussion.

**Maximum-a-posteriori (MAP) estimation.** Suppose there are a set of finite classes $\mathcal{C}$. A class label $Y \in \mathcal{C}$ is generated with probability $\pi_Y$, where $\sum_{Y \in \mathcal{C}} \pi_Y = 1$. Given $Y$, the corresponding feature $X$ in the space $\mathcal{X}$ is generated from the distribution $X \sim \mathbb{P}_Y$. A classifier is a decision $f : \mathcal{X} \to \mathcal{C}$ and the *Bayesian mis-classification error* can be characterized as $\xi(f) = \sum_{Y \in \mathcal{C}} \pi_Y \int 1_{f(X) \neq Y} \mathbb{P}_Y(X)$, where and later $1_S$ indicates 1 if $S$ is true and 0 otherwise. The *MAP estimation* of $Y$ given $X$ is the classifier $f^*(X) \triangleq \arg \max_{Y \in \mathcal{C}} \pi_Y \mathbb{P}_Y(X)$ that can minimize $\xi(f)$ [35]. Later, we denote the minimal Bayesian mis-classification error $\xi(f)$ as $\xi^* = \xi(f^*)$.

**Signal-to-Noise Ratio (SNR).** Detection of a signal from the background essentially corresponds to a binary classification problem. SNR is widely used to measure the potential detection performance before specifying the classifier [81]. In particular, if we have two equiprobable classes $\mathcal{C} = \{-1, 1\}$ and the features follows 1-d Gaussian distributions $\mathbb{P}_Y = \mathcal{N}(\mu_Y, \sigma^2), Y \in \mathcal{C}$. The SNR $\rho$ defined as follows precisely characterizes the minimal Bayesian mis-classification error.

$$\textbf{SNR:} \quad \rho = \frac{\text{mean difference}^2}{\text{variance}} = \frac{(\mu_1 - \mu_{-1})^2}{\sigma^2}. \tag{1}$$

In this case, the MAP estimation $f^*(X) = 2 * 1_{|X-\mu_1| \geq |X-\mu_{-1}|} - 1$ and the minimal Bayesian mis-classification error is $\Phi(-\sqrt{\rho}/2)$ where $\Phi$ denotes the cumulative standard Gaussian distribution function. For more general cases where the two classes are associated with sub-Guassian distributions $\mathbb{P}_Y$, s.t. $\mathbb{P}_Y(|X| > t) \in [c_1 \exp(-c_2 t^2), C_1 \exp(-C_2 t^2)]$, for some non-negative constants $c_1, c_2, C_1, C_2$, a similar connection between $\xi^*$ and $\rho$ can be shown by leveraging sharp sub-Gaussian lower bounds [82]. We will specify the connection to SNR in our case in Sec. 4 and the SNR $\rho$ will be used as the main bridge to compare the mis-classification errors of non-linear v.s. linear models.

**Contextual Stochastic Block Model (CSBM).** Random graph models have been widely used to study the performance of algorithms on graphs [83, 84]. For node classification problems, CSBM is often used [68–70], as it well combines the models of network structure and node attributes.

We study the case that nodes are in two equi-probable classes $\mathcal{C} = \{-1, 1\}$, where $\pi_Y = \frac{1}{2}, Y \in \mathcal{C}$. Our analysis can be generalized. An attributed network $\mathcal{G} = (\mathcal{V}, \mathcal{E}, \mathbf{X})$ is sampled from CSBM with parameters $(n, p, q, \mathbb{P}_1, \mathbb{P}_{-1})$ as follows. Suppose there are $n$ nodes, $\mathcal{V} = [n]$. For each node $v$, the label $Y_v \in \mathcal{C}$ is sampled from Rademacher distribution. Given $Y_v$, the node attribute $X_v$ is sampled from $\mathbb{P}_{Y_v}$. For two nodes $u, v$, if $Y_u = Y_v$, there is an edge $e_{uv} \in \mathcal{E}$ connecting them with probability $p$. If $Y_u \neq Y_v$, there is an edge $e_{uv} \in \mathcal{E}$ connecting them with probability $q$. All node attributes $\mathbf{X}$ and edges $\mathcal{E}$ are independent given the node labels $\mathbf{Y} = \{Y_v | v \in \mathcal{V}\}$.

Note that $p > q$ indicates the nodes with the same labels tend to be directly connected, which corresponds to the *homophilic* case, while $p < q$ corresponds to the *heterophilic* case.

The gap $|p - q|$, representing probabilities difference of a node connects to nodes from the same class or the different class, reflects *structural information* and the gap between $\mathbb{P}_1$, $\mathbb{P}_{-1}$ reflects *attributed information*, e.g., Jensen-Shannon distance $\text{JS}(\mathbb{P}_1, \mathbb{P}_{-1})$ that is well connected to Bayesian mis-classification error [85]. Graph learning allows combining these two types of information. In Sec. 4, we give more specific definitions of these two types of information and their regime for our analysis.

## 3 Bayesian Inference and Nonlinearity in Graph Neural Networks

In the previous section, we discuss that given conditioned feature distributions $X \sim \mathbb{P}_Y, Y \in \mathcal{C}$, the MAP estimation $f^*(X)$ can minimize mis-classification error. For node classification in an attributed network, the estimation of a node label should depend on not only one's own attributes but also its neighbors'. For example, in a homophilic network, nodes with same labels tend to be directly connected. Intuitively, using the averaged neighbor attributes may provide better estimation of the label, which gives us graph convolution. In a heterophilic network, nodes with different labels tend to be directed connected. So, intuitively, checking the difference between one's attributes and the neighbors' may provide better estimation. However, what could be the optimal form to combine one's own attribute with the neighbors' attributes? We resort to the MAP estimation. That is, given the attributes of a node $v \in \mathcal{V}$ and its neighbors $\mathcal{N}_v$, we consider the MAP estimation as follows.

$$f^*(X_v, \{X_u\}_{u \in \mathcal{N}_v}) = \underset{Y_v \in \mathcal{C}}{\arg\max} \underset{Y_u \in \mathcal{C}, \forall u \in \mathcal{N}_v}{\max} \pi_{Y_v, \{Y_u\}_{u \in \mathcal{N}_v}} \mathbb{P}\left(X_v, \{X_u\}_{u \in \mathcal{N}_v}, \mathcal{N}_v | Y_v, \{Y_u\}_{u \in \mathcal{N}_v}\right),$$

where $\pi_{Y_v, \{Y_u\}_{u \in \mathcal{N}_v}}$ denotes their prior distributions of node labels. Note that here we simplify the problem and consider only 1-hop neighbors by following the setting [71]. In practice, most GNN models can only work on local networks due to the scalability constraints [11, 86, 87]. Even with the above simplification, the above MAP estimation is generally intractable.

Therefore, we consider the CSBM with parameters $(n, p, q, \mathbb{P}_1, \mathbb{P}_{-1})$. In this case, the prior distribution follows $\pi_{Y_v, \{Y_u\}_{u \in \mathcal{N}_v}} = 2^{-|\mathcal{N}_v|-1}$, which is a constant given $\mathcal{N}_v$. The rest term follows

$$\mathbb{P}\left(X_v, \{X_u\}_{u \in \mathcal{N}_v}, \mathcal{N}_v | Y_v, \{Y_u\}_{u \in \mathcal{N}_v}\right) = \mathbb{P}\left(X_v, \{X_u\}_{u \in \mathcal{N}_v} | Y_v, \{Y_u\}_{u \in \mathcal{N}_v}\right) \mathbb{P}\left(\mathcal{N}_v | Y_v, \{Y_u\}_{u \in \mathcal{N}_v}\right)$$

$$= \prod_{u \in \mathcal{N}_v \cup \{v\}} \mathbb{P}_{Y_u}(X_u) \prod_{u \in \mathcal{N}_v} p^{(1+Y_vY_u)/2} q^{(1-Y_vY_u)/2} \quad (2)$$

Therefore, the MAP estimation $f^*(X_v, \{X_u\}_{u \in \mathcal{N}_v})$ is to solve

$$f^*(X_v, \{X_u\}_{u \in \mathcal{N}_v}) = \underset{Y_v \in \mathcal{C}}{\arg\max} \; \mathbb{P}_{Y_v}(X_v) \prod_{u \in \mathcal{N}_v} \underset{Y_u \in \mathcal{C}}{\max} \; \mathbb{P}_{Y_u}(X_u) p^{(1+Y_vY_u)/2} q^{(1-Y_vY_u)/2} \quad (3)$$

This can be solved via the max-product algorithm [88]. To establish the connection to GNNs, we rewrite the RHS of Eq. 3 in the logarithmic form and use the fact that $\mathcal{C} = \{-1, 1\}$. And, we achieve

$$f^*(X_v, \{X_u\}_{u \in \mathcal{N}_v}) = \text{sgn}\left(\log \frac{\mathbb{P}_1(X_v)}{\mathbb{P}_{-1}(X_v)} + \sum_{u \in \mathcal{N}_v} \mathcal{M}(X_u, p, q)\right), \quad \text{where}$$

$$\mathcal{M}(X_u, p, q) = \text{ReLU}\left(\log \frac{\mathbb{P}_1(X_v)}{\mathbb{P}_{-1}(X_v)} + \log \frac{p}{q}\right) - \text{ReLU}\left(\log \frac{\mathbb{P}_1(X_v)}{\mathbb{P}_{-1}(X_v)} + \log \frac{q}{p}\right) + \log \frac{q}{p}.$$

We leave the derivation in Appendix B. Amazingly, activation ReLUs in the message $\mathcal{M}$ well connect to the activations commonly-used in GNN models, e.g., graph convolution networks [12]. Given the optimality of the MAP estimation, we summarize this observation in Proposition 1.

**Proposition 1** (**Optimal Nonlinear Propagation**). *Consider a network $\mathcal{G} \sim CSBM(n, p, q, \mathbb{P}_1, \mathbb{P}_{-1})$. To classify a node $v$, the optimal nonlinear propagation (derived by the MAP estimation) given the attributes of $v$ and its neighbors follows:*

$$\mathcal{P}_v = \psi(X_v; \mathbb{P}_1, \mathbb{P}_{-1}) + \sum_{u \in \mathcal{N}_v} \phi(\psi(X_u; \mathbb{P}_1, \mathbb{P}_{-1}); \log(p/q)) \quad (4)$$

*where $\psi(a; \mathbb{P}_1, \mathbb{P}_{-1}) = \log \frac{\mathbb{P}_1(a)}{\mathbb{P}_{-1}(a)}$ and $\phi(a; \log \frac{p}{q}) = ReLU(a + \log \frac{p}{q}) - ReLU(a - \log \frac{p}{q}) - \log \frac{p}{q}$.*

The optimal nonlinear propagation in Eq. (4) may contain two types of non-linear functions: (1) $\psi$ is to measure the likelihood ratio between two classes given the node attributes; (2) $\phi$ is to propagate the likelihood ratios of the neighbors. ReLUs in $\phi$ avoid the overuse of the likelihood ratios from neighbors, as $\phi$ essentially provides a bounded function (See Fig. 2). One observation of the direct benefit of this non-linear propagation is as follows.

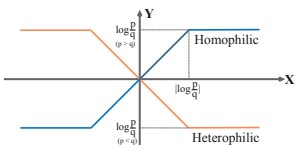

Figure 2: Function $\phi(x; \log \frac{p}{q})$.

**Remark 1.** *When there is no structural information, i.e., $p = q$, $\phi(x; 0) = 0$, $\forall x \in \mathbb{R}$, the propagation is deactivated, which avoids potential contamination from the attributes of the neighbors.*

In the equiprobable case, the MAP estimation also gives the maximum likelihood estimation (MLE) of $Y$ if we view the labels as the fixed parameters. When the classes are unbalanced $\pi_Y \neq \frac{1}{2}$, similar results can be obtained while additional terms $\log \frac{\pi_1}{\pi_{-1}}$ may appear as bias in Eq. (4). Later, our analysis focuses on the equiprobable case while empirical results in Sec. 5 show more general cases.

Moreover, if one is to infer the posterior distribution of $Y$, one may replace the max-product algorithm to solve Eq. (3) with the sum-product algorithm [89]. Then, the obtained non-linearity in $\phi$ will turn into Tanh functions. As ReLUs are more used in practical GNNs, we focus on the case with ReLUs.

**Discussion on the Non-linearity.** Next, we discuss more insights into the non-linearity of $\psi$ and $\phi$.

The function $\psi$ essentially corresponds to *a node-attribute transformation*, which depends on the distributions $\mathbb{P}_{\pm 1}$. As these distributions are unknown in practice, a NN model to learn $\psi$ is suggested, such as the one in the model APPNP [21] and GPR-GNN [25]. Due to the expressivity of NNs [90, 91], a wide range of $\psi$ can be modeled. One insightful example is that when $\mathbb{P}_{\pm 1}$ are Laplace distributions, $\psi$ is a bounded function (same as $\phi$) to control the heavy-tailed attributes generated by Laplace distributions.

**Example 1** (Laplace Assumption). *When node attributes follow $m$-dimensional independent Laplace distribution, i.e., $\mathbb{P}_{Y_v} = \frac{1}{(2b)^m} \exp(-\|X_v - Y_v \mu\|_1 / b)$ for $\mu \in \mathbb{R}^m$, $b > 0$ and $Y_v \in \{-1, 1\}$. According to Eq. (4), the function $\psi(\cdot; \mathbb{P}_1, \mathbb{P}_{-1})$ can be specified as*

$\psi_{lap}(X_v; \mathbb{P}_1, \mathbb{P}_{-1}) = \mathbb{1}^T \phi(X_v; 2\mu/b)$, *where $\phi$ as defined in Eq. (4) works in an entry-wise manner.*

As node-attribute distributions may vary a lot, $\psi$ is better to be modeled via a general NN in practice. More interesting findings may come from $\phi$ in Eq. (4) as it has a fixed form and well matches the most commonly-used GNN architecture. Specifically, besides the extreme case stated in Remark 1, we are to investigate *how* non-linearity induced by the ReLUs in $\phi$ may benefit the model. We expect the findings to provide the explanations to some previous empirical experiences on using GNNs.

To simplify our discussion, when analyzing $\phi$, we focus on the case with a linear node-attribute transformation $\psi = \psi_{\text{Gau}}$ in Eq. (6) by assuming label-dependent Gaussian node attributes. This follows the assumptions in previous studies [71, 72]. In fact, there are a class of distributions named natural exponential family (NEF) [34] which if the node attributes satisfy, the induced $\phi$ is linear. We conjecture that our later results in Sec. 4 are applied to the general NEF since the only difference is the bias term by comparing Eq. (5) and Eq. (6).

**Example 2** (Natural Exponential Family Assumption). *When node features follow $m$-dimensional natural exponential family distributions $\mathbb{P}_{Y_v}(X) = h(X_v) \cdot \exp(\theta_{Y_v}^T X_v - M(\theta_{Y_v}))$ for $\theta_{Y_v} \in \mathbb{R}^m$ and $Y_v \in \{-1, 1\}$ where $M(\theta_{Y_v})$ is a parameter function. The function $\psi(\cdot; \mathbb{P}_1, \mathbb{P}_{-1})$ is specified as:*

$$\psi_{nef}(X_v; \theta_1, \theta_{-1}) = (\theta_1 - \theta_{-1})^T X_v - (M(\theta_1) - M(\theta_{-1})). \tag{5}$$

*In particular, when $\mathbb{P}_1 = \mathcal{N}(\mu, I/m)$, $\mathbb{P}_{-1} = \mathcal{N}(\nu, I/m)$ for $\mu, \nu \in \mathbb{R}^m$,*

$$\psi_{Gau}(X_v; \mu, \nu) = m \left[ (\mu - \nu)^T X_v - (\|\mu\|_2^2 - \|\nu\|_2^2)/2 \right]. \tag{6}$$

More generally, our optimal nonlinear propagation Eq. (4) can be well generalized to other settings as long as the model satisfies *edge-independent assumption*, where edges random variables are mutually independent conditioned on the labels of nodes. When this assumption is satisfied, the MAP estimation will result in graph convolution with ReLU activation.

We summarize our main theoretical findings regarding the nonlinearity of $\phi$ in the next section.

## 4 Main Results on ReLU-based Nonlinear Propagation

In this section, we summarize our analytical results on $\phi$ in the optimal nonlinear propagation (Eq. (4)). Our study assumes an attributed network generated from $\mathrm{CSBM}(n, p, q, \mathcal{N}(\mu, I/m), \mathcal{N}(\nu, I/m))$ where $\mu, \nu \in \mathbb{R}^m$. We use CSBM-G$(n, p, q, \mu, \nu)$ later to denote this model for simplicity. We are interested in the asymptotic behavior when $n \to \infty$. Note that all parameters $\mu, \nu, p, q, m$ may implicitly depend on $n$. We are to compare the non-linear propagation model $\mathcal{P}_v$ suggested by Eq. (4) where $\psi = \psi_{\mathrm{Gau}}$ with the following linear counterpart $\mathcal{P}_v^l$.

**Baseline linear model:** $\quad \mathcal{P}_v^l(w) = \psi_{\mathrm{Gau}}(X_v; \mu, \nu) + w \sum_{u \in \mathcal{N}_v} \psi_{\mathrm{Gau}}(X_u; \mu, \nu), \text{ for all } v \in \mathcal{V}.$ (7)

where $w \in \mathbb{R}$ is an extra parameter to be tuned. Note that this linear model can be claimed as an optimal linear model up-to a choice of $w$ because the distributions of both the center node attribute $X_v$ and the linear aggregation from the neighbors $\sum_{u \in \mathcal{N}_v} X_u$ are Gaussian and symmetric w.r.t. the hyperplane $\{Z \in \mathbb{R}^m | (\mu - \nu)^T Z = (\|\mu\|_2^2 - \|\nu\|_2^2)/2\}$ for the two classes. We are to compare their classification errors $\xi^r = \xi(\mathrm{sgn}(\mathcal{P}_v))$ and $\xi^l(w) = \xi(\mathrm{sgn}(\mathcal{P}_v^l(w)))$. By following [71], we also discuss separability of all nodes in the network, i.e., $\mathbb{P}(\forall v \in \mathcal{V}, \mathcal{P}_v \cdot Y_v > 0)$ in Theorem 1.

To begin with, we introduce several quantities for the convenience of further statements. The SNRs

$$\rho_r = \frac{(\mathbb{E}[\mathcal{P}_v | Y_v = 1] - \mathbb{E}[\mathcal{P}_v | Y_v = -1])^2}{\mathrm{var}(\mathcal{P}_v | Y_v = 1)}, \rho_l(w) = \frac{(\mathbb{E}[\mathcal{P}_v^l(w) | Y_v = 1] - \mathbb{E}[\mathcal{P}_v^l(w) | Y_v = -1])^2}{\mathrm{var}(\mathcal{P}_v^l(w) | Y_v = 1)}$$

are important quantities to later characterize different types of propagation. Also, we characterize structural information by $\mathcal{S}(p, q) = (p - q)^2/(p + q)$ and attributed information by $\sqrt{m}\|\mu - \nu\|_2$.

**Assumption 1** (Moderate Structural Information). $\mathcal{S}(p, q) = \omega_n(\frac{(\log n)^2}{n})$ and $\frac{\mathcal{S}(p,q)}{|p-q|} \nrightarrow 1$.

**Assumption 2** (Bounded Attributed Information). $\sqrt{m}\|\mu - \nu\|_2 = o_n(\log n)$.

Assumption 1 states that structural information should be neither too weak nor too strong. $\mathcal{S}(p, q) = \omega_n(\frac{(\log n)^2}{n})$ excludes the extremely weak case discussed in Remark 1. Moreover, the graph structure should not be too sparse, so the aggregated information from neighbors dominates the propagation. $\frac{\mathcal{S}(p,q)}{|p-q|} \nrightarrow 1$ means neither $p = \omega_n(q)$ nor $q = \omega_n(p)$, which avoids extremely strong structural information. This assumption is more general than some concurrent works on CSBM-based GNN analysis [71, 72] as we include the cases with less structural information $|p - q| = o_n(p + q)$ and with heterophily $p < q$. Assumption 2 is to avoid too strong attributed information: when $\sqrt{m}\|\mu - \nu\|_2 = \Omega_n(\log n)$, all nodes in CSBM can be accurately classified in the asymptotic sense without structural information, i.e. $\mathbb{P}(\forall v \in \mathcal{V}, \psi_{\mathrm{Gau}}(X_v; \mu, \nu) \cdot Y_v > 0) = 1 - o_n(1)$. Now, we present our first lemma which links the mis-classification errors $\xi^r, \xi^l$ to with the SNRs $\rho_r, \rho_l$:

**Lemma 1.** *Suppose $(p, q)$ satisfies Assumption 1, for any $\mathcal{G} \sim$ CSBM-G$(n, p, q, \mu, \nu)$,*

$$\xi^r \in [C_1 \exp(-C_2 \rho_r/2), \exp(-\rho_r/2)], \quad \xi^l(w) \to \exp(-\rho_l(w)(1 + o_n(1))/2) \quad (8)$$

where $C_2$ is asymptotically a constant, and the notation $a(n) \to b(n)$ denotes $a(n)/b(n) \to 1$. Lemma 1 claims that the classification errors under both nonlinear and linear model can be controlled by their SNRs. By leveraging Lemma 1, we can further illustrate the separability of all nodes in the network, which is presented in the following theorem.

**Theorem 1** (Separability). *Suppose that $(p, q)$ satisfies Assumption 1 for $\mathcal{G} \sim$ CSBM-G$(n, p, q, \mu, \nu)$, if $\sqrt{m}\|\mu - \nu\|_2 = \omega_n(\sqrt{\log n/\mathcal{S}(p,q)n})$, then*

$$\mathbb{P}(\forall v \in \mathcal{V}, \mathcal{P}_v \cdot Y_v > 0) = 1 - \mathcal{O}_n(n \exp(-\rho_r/2)) = 1 - o_n(1), \quad (9)$$

$$\mathbb{P}(\forall v \in \mathcal{V}, \mathcal{P}_v^l(w) \cdot Y_v > 0) = 1 - \mathcal{O}_n(n \exp(-\rho_l(w)/2)) = 1 - o_n(1). \quad (10)$$

*Here, assume $|w| > c$ for some positive constant $c$ and $sgn(w) = sgn(p - q)$ in the linear model.*

Theorem 1 applies to both homophilic ($p > q$) and heterophilic ($p < q$) scenarios. Even for just the linear case, compared to [71] which needs $\sqrt{m}\|\mu - \nu\|_2 = \omega_n(\log n/\sqrt{\mathcal{S}(p,q)n})$ to achieve separability, we have $\sqrt{\log n}$ improvement due to a tight analysis.

As shown in Lemma 1 and Theorem 1, the errors are mainly determined by SNRs. Large SNR implies a fast decay rate of the errors of a single node and the entire graph, which motivates us to further explore SNRs to illustrate a comparison between non-linear and linear models. We consider comparing with the optimal linear model, i.e., $\rho_l^* = \rho_l(w^*)$, where $w^* = \arg\min_{w \in \mathbb{R}} \xi^l(w)$.

**Theorem 2.** *Suppose that $(p, q)$ satisfies Assumption 1, for $\mathcal{G} \sim CSBM\text{-}G(n, p, q, \mu, \nu)$, under the separable condition in Theorem 1 $\sqrt{m}\|\mu - \nu\|_2 = \omega_n(\sqrt{\log n / S(p,q)n})$, we further have*

- *I. Limited Attributed Information: When $\sqrt{m}\|\mu - \nu\|_2 = \mathcal{O}_n(1)$,*

$$\rho_r = \Theta_n(\rho_l^*), \tag{11}$$

  *Further, if $\sqrt{m}\|\mu - \nu\|_2 = o_n(|\log(p/q)|)$, $\rho_r / \rho_l^* \to 1$;*
- *II. Sufficient Attributed Information: When $\sqrt{m}\|\mu - \nu\|_2 = \omega_n(1)$ and satisfies Assumption 2,*

$$\rho_r = \omega_n(\min\{\exp(m\|\mu - \nu\|_2^2/3), nS(p,q)m^{-1}\|\mu - \nu\|_2^{-2}\} \cdot \rho_l^*) = \omega_n(\rho_l^*). \tag{12}$$

Theorem 2 also works both homophilic ($p > q$) and heterophilic ($p < q$) scenarios. Theorem 2 implies that when attributed information is limited, nonlinear propagation behaves similar to the linear model as their SNRs are in the same order. Particularly, when attributed information is very limited, $\sqrt{m}\|\mu - \nu\|_2 = o_n(|\log(p/q)|)$, the SNRs of two models are asymptotically the same. In the regime of sufficient attributed information, nonlinear propagation brings order-level superiority compared with the linear model. The intuition is that in this regime, when the attributes are very informative, the bounds of $\phi$ in Eq. (4) help with avoiding overconfidence given by the node attributes. The coefficient before $\rho_l^*$ in Eq. (12) shows the trade-off between structural information and attributed information on controlling the superiority of nonlinear propagation.

Next, we analyze whether nonlinearity makes model more transferable or not when there often exists a distribution shift between the training and testing datasets, which is also practically useful.

We consider the following setting. We assume using a large enough network generated by CSBM-G$(n, p, q, \mu, \nu)$ for training so that the optimal parameters as in $\mathcal{P}_v$ and $\mathcal{P}_v^l(w^*)$ for this CSBM-G have been learnt. We consider their mis-classification errors over another CSBM-G with parameters $(n, p', q', \mu', \nu')$. We keep the amounts of attributed information and structural information unchanged by setting $p = p'$, $q = q'$, $\mu' = (\mu + \nu)/2 + \mathbf{R}(\mu - \nu)/2$, $\nu' = \mu + \nu/2 + \mathbf{R}(\nu - \mu)/2$ for a rotation matrix $\mathbf{R}$ close to $I$. Let $\Delta\xi^r$ and $\Delta\xi^l(w^*)$ denote the increase of mis-classification errors of models $\mathcal{P}_v$ and $\mathcal{P}_v^l(w^*)$, respectively, due to such a distribution shift. We may achieve the following results.

**Theorem 3** (Transferability). *Suppose that $(p, q)$ satisfies Assumption 1, for $\mathcal{G}' \sim CSBM\text{-}G(n, p', q', \mu', \nu')$, under the linear separable condition $\sqrt{m}\|\mu' - \nu'\|_2 = \omega_n(\sqrt{\log n / S(p', q')n})$. Suppose $\mathcal{P}_v$ and $\mathcal{P}_v^l(w^*)$ have learnt parameters from $\mathcal{G} \sim CSBM\text{-}G(n, p, q, \mu, \nu)$ where the parameters of two CSBM-Gs follow the relation described above. Then, we have*

- *I. Limited Attributed Information: When $\sqrt{m}\|\mu - \nu\|_2 = o_n(|\log(p/q)|)$, $\Delta\xi^r / \Delta\xi^l(w^*) \to 1$.*
- *II. Sufficient Attributed Information: When $\sqrt{m}\|\mu - \nu\|_2 = \omega_n(1)$ and and satisfies Assumption 2, $\Delta\xi^r / \Delta\xi^l(w^*) \to 0$.*

Similar to Theorem 2, when attributed information is very limited, nonlinearity will not bring any benefit, while in the regime with informative attributes, nonlinearity increases model transferability. We leave the intermediate regime $\sqrt{m}\|\mu - \nu\|_2 \in [\Omega_n(|\log(p/q)|), O_n(1)]$ for future study.

## 5 Experiments

In this section, we verify our theoretical results based on synthetic and real datasets. In all experiments, we fix $w$ in the linear model (Eq. (7)) as $w = 1$ for the homophilic case ($p > q$) and $w = -1$ for the heterophilic case ($p < q$). Experiments on other $w$'s can be found in Appendix H.1.1, which does not change the achieved conclusion. This is because when the node number $n$ is large, for a constant $w$, the neighbor information will dominate the results. Later, we use $\mathcal{P}_v^l = \mathcal{P}_v^l(w)$ for simplicity.

### 5.1 Asymptotic Experiments - Model Accuracy & Transferability Study

Our first experiments focus on evaluating the asymptotic ($n \to \infty$) classification performance of nonlinear and linear models. Given a CSBM-G, we generate 5 graphs and compute the average

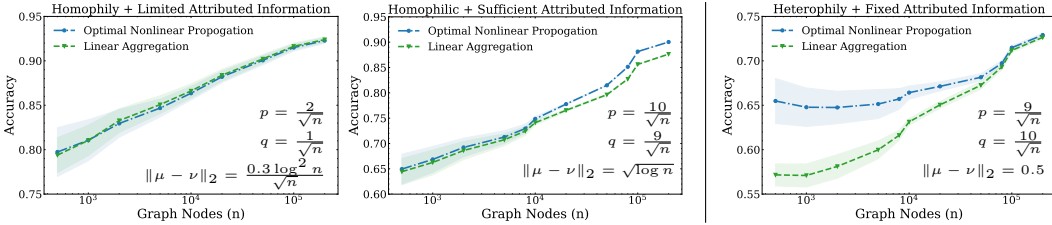

Figure 3: Classification Performance on Nonlinear Models v.s. Linear Models ($\mathcal{P}_v$ v.s. $\mathcal{P}_v^l$). LEFT: Homophily + Limited Attr. Info.; MIDDLE: Homophily + Suff. Attr. Info.; RIGHT: Heterophily + Fixed Attr. Info.. $m = 10$ and other parameters are listed in the figures.

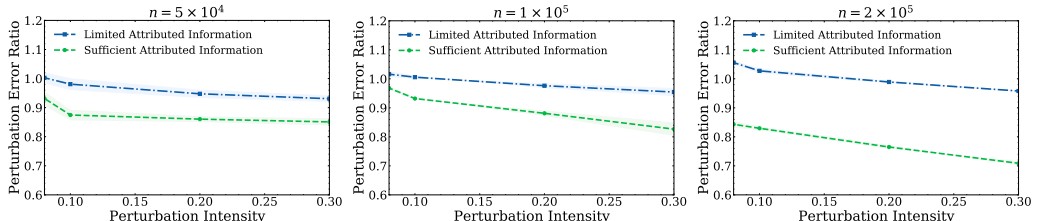

Figure 4: Perturbation Intensity ($1 - \langle \mu' - \nu', \mu - \nu \rangle / \|\mu - \nu\|_2^2$) v.s. Perturbation Error Ratio ($\Delta \xi^r / \Delta \xi^l$) with Different Node Numbers. Other parameters are: $p = 2\sqrt{n}/n, q = \sqrt{n}/n$; Limited Attr. Info. $\|\mu - \nu\|_2 = 0.3 \log^2 n/\sqrt{n}$; Suff. Attr. Info. $\|\mu - \nu\|_2 = 0.1\sqrt{\log n}$.

accuracy results (#correctly classified nodes / #total nodes). We compare the nonlinear v.s. linear models under three different CSBM-G settings. Fig. 3 shows the results.

All three cases satisfy the separability condition in Theorem 1, so, as $n$ increases, the accuracy progressively increases to 1. Our results also match well with the implications provided by Theorem 2. In the regime with limited attribute information (Fig. 1 LEFT) where $\rho_r = \Theta_n(\rho_l^*)$ as proved, the nonlinear model and the linear model behave almost the same (performance gap $< 0.15\%$ for $n \geq 10^5$). In the regime with sufficient attribute information (Fig. 1 MIDDLE) where $\rho_r = \omega_n(\rho_l^*)$ as proved, we may observe that the nonlinear model can significantly outperform the linear model as $n \to \infty$. Fig. 1 RIGHT is to show the heterophilic graph case ($p < q$). If we switch the values of $p, q$ (and also change the models correspondingly), we obtain the exactly same figure up to some experimental randomness (see Appendix H.1.2). Also, Fig. 1 RIGHT considers a boundary case of sufficient attributed information, i.e., $\sqrt{m}\|\mu - \nu\|_2 = \Theta_n(1)$. We observe that Theorem 2 still well describes the asymptotic performance when $n \to \infty$.

We further study the transferability for the non-linear model and the linear model. We follow the setting in Theorem 3 by rotating $\mu, \nu \to \mu', \nu'$. Fig. 4 shows the result and well matches Theorem 3. In the regime of limited attributed information, the two models have the almost same transferability, i.e., the perturbation error ratio is close to 1. In contrast, with sufficient attributed information, the non-linear model is more transferrable than the linear counterpart as the ratio is smaller than 1.

## 5.2 Transition Curve

Our second experiment studies the tradeoff between attributed information and structural information. We fix the graph size $n = 2 \times 10^4$ and get the averaged classification accuracy based on 5 generated graphs. For the homophilic case, we test different levels of attributed information ($\|\mu - \nu\|$ from $10^{-4}$ to 10 with $m = 10$) and structural information (fixing $q = 5 \times 10^{-3}$ and increasing $p$ from $p = q$ to 1). The intermediate testing points are sampled in log scales. Fig. 5 LEFT shows the results. When structural information is limited and attributed information is sufficient, the non-linear model shows significant advantage over the linear model while for most other parameter settings, these two models share similar performance. Fig. 5 RIGHT shows the heterophilic case, where we observe a similar pattern. In the heterophilic case, we fixing $p = 5 \times 10^{-3}$ and increasing $q$ from $q = p$ to 1.

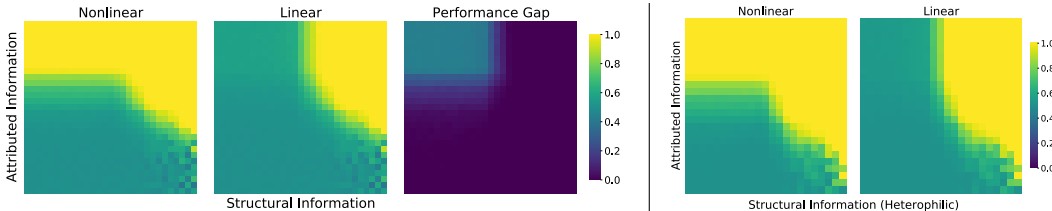

Figure 5: Transition Curves Attributed Information ($\sqrt{m}\|\mu - \nu\|_2$) v.s. Structural Information ($|\log(p/q)|$) for CSBM-G with Homophilic (LEFT) / Heterophilic (RIGHT) Graph Structures. The values in Performance Gap are obtained by the nonlinear case subtracting the linear case.

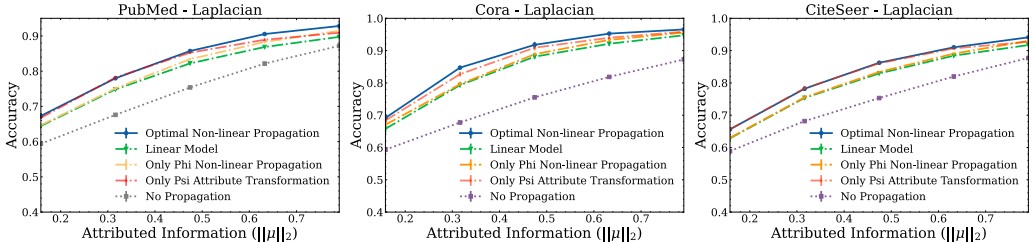

Figure 6: Averaged one-vs-all Classification Accuracies on Citation Networks of Different Nonlinear Models v.s. Linear Models. Node attributes in or out of the one class are generated from Laplace distributions with different means $\pm\mu$ and $b = 1$ (Example 1). The optimal non-linear model has advantage over the models with only nonlinear attribute transformation ($\psi_{\text{lap}}$), with only nonlinear information propagation ($\phi$), the linear model.

## 5.3 Real-world Network Experiments

This experiments compare non-linear models and linear models under Gaussian and Laplacian attributes on three benchmark citation networks PubMed, Cora, and CiteSeer [92]. In these three networks, nodes denote papers and edges denote the citation relationships between the papers. The statistics (# nodes, # edges, # classes) of these three networks are: PubMed (19,717, 44,338, 3); Cora (2,708, 5,428, 7); CiteSeer (3,327, 4,732, 6).

**Experimental Settings.** We carry out one-v.s.-all and several-v.s.-several classification tasks. After nodes are put into two classes, we generate two graphs independently with attributes according to Gaussian (or Laplace) distributions. One graph is used for training and the other one for testing. For the Gaussian case, we use a nonlinear model by following Eq. 4 with $\psi = \psi_{\text{Gau}}$ while the parameters such as $\log(p/q)$, $\mu - \nu$ and other biases need to be learned. For the Laplacian case, we consider three nonlinear models by following the form of (a) full Eq. 4 with $\psi = \psi_{\text{lap}}$; (b) only nonlinear attribute transformation $\psi = \psi_{\text{lap}}$; (c) only nonlinear propagation $\phi$ with linear attribute transformation. Later, we call them nonlinear models (a), (b), (c), respectively. Similar to the Gaussian case, all the parameters in these functions are obtained by training. The model is trained with Adam optimizer (learning rate = $1e - 2$, weight decay = $5e - 4$). We give other details to Appendix H.2.1.

**Result Analysis.** We report the averaged results over 5 trials in Fig. 1 (Gaussian) and Fig. 6 (Laplacian). Due to the space limit, we leave the results for the several-v.s.-several case in Appendix H.2.1. The Gaussian case well matches our theory. Only when the node features are very informative, the gaps between the nonlinear model and the linear model become significant. This is true for all three networks.

The Laplacian case is more complicated. Non-linear model (a) outperforms the two non-linear models (b) and (c). The two non-linear models both outperform the linear model. More specifically, when attributed information is not very informative, i.e., small $\|\mu\|_2$, attribute nonlinear transformation function $\psi_{\text{Lap}}$ is more crucial, because in this regime, non-linear model (a) significantly outperforms non-linear model (c) and non-linear model (b) significantly outperforms the linear model, while two non-linear models (a) and (b) perform similarly, and non-linear model (c) and the linear model perform similarly. With more informative attributed information, nonlinear propagation function $\phi$ becomes more significant, because the gaps between two non-linear models (a) and (b) (also,

non-linear model (c) and the linear model) are obvious, which again matches our Theorem 2 although here we have Laplacian node attributes instead of Gaussian node attributes.

## 6 Conclusion

This work uses Bayesian methods to investigate the function of non-linearity in GNNs. Given a graph generated from CSBM, we observe the optimal non-linearity to estimate a node label given its own and neighbors' attributes is in twofold: attribute non-linear transformation and non-linear propagation. We further investigate the non-linear propagation by imposing Gaussian assumptions on node attributes. We prove that non-linear propagation shares a similar performance (with or without distribution shift) with linear propagation in most cases except when node attributes become very informative. These findings explain many previous empirical observations in this domain and would help researchers and practitioners to understand their GNNs' behaviors in practice.

## 7 Acknowledgement

We greatly thank all the reviewers for valuable feedback and actionable suggestions. R. Wei, H. Yin and P. Li are partially supported by 2021 JPMorgan Faculty Award and NSF award OAC-2117997.

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
