# OpenReview forum: "Understanding Non-linearity in Graph Neural Networks from the Bayesian-Inference Perspective"
_NeurIPS.cc/2022/Conference — NeurIPS 2022 Accept_

### Official Review · Reviewer_o4Ni · 2022-06-27

**Rating:** 3
**Confidence:** 3
**Soundness:** 2 fair
**Presentation:** 2 fair
**Contribution:** 2 fair

**Summary:**

This paper studies when non-linearity is useful for accuracy in graph neural nets. Authors show that under assumptions, the superiority of ReLU activations is only significant when the node attributes are far more informative than the graph structure.
----
Authors say that I gave "accept score" initially. This is wrong.
Somehow this year "5" means borderline accept, but usually it meant "weak reject". So I intended to reject. I took another look of the paper and I feel the paper is not ready for publication so changed score to reject.

**Questions:**

1. The main conclusion is vague and not so clear. The main point of the paper is trying to claim that "the superiority of ReLU activations is only significant when the node attributes are far more informative than the graph structure." However, it's not clear what you mean by node attributes are more informative than graph strucutre. A very clear definition and intuition, examples need to be give.
2. It's not clear what practical implication we can gain? Why not just always add non-linearity, since they will outperform in some cases?

Authors need to clearly resolve my concerns to raise rating.

**Limitations:**

1. The main conclusion is vague and not so clear. The main point of the paper is trying to claim that "the superiority of ReLU activations is only significant when the node attributes are far more informative than the graph structure." However, it's not clear what you mean by node attributes are more informative than graph strucutre. A very clear definition and intuition, examples need to be give.
2. It's not clear what practical implication we can gain? Why not just always add non-linearity, since they will outperform in some cases?

Authors need to clearly resolve my concerns to raise rating.

**Strengths And Weaknesses:**

Strength:
1. The problem of non-linearity is interesting and useful in practice.
2. The result shows some interesting implications, e.g., when structure vs feature is more important. Though the main message is too vague to be useful.

Weakness:
1. The main conclusion is vague and not so clear. The main point of the paper is trying to claim that "the superiority of ReLU activations is only significant when the node attributes are far more informative than the graph structure." However, it's not clear what you mean by node attributes are more informative than graph strucutre. A very clear definition and intuition, examples need to be give.
2. It's not clear what practical implication we can gain? Why not just always add non-linearity, since they will outperform in some cases?

Authors need to clearly resolve my concerns to raise rating.

---

> ### Author Response · Authors · 2022-07-31
> **Response to Reviewer o4Ni**
>
> We thank Reviewer o4Ni for reviewing our work. We note that the reviewer initially gave an acceptance recommendation (score 5, before Jul. 30th) and then changed it to a reject recommendation (score 3) while no further reasons were provided. In the following, we first address the questions currently raised by the reviewer. We sincerely would like to know if there are any other concerns that make the reviewer change the recommendation.
>
> > "The main conclusion is vague and not so clear. The main point of the paper is trying to claim that "the superiority of ReLU activations is only significant when the node attributes are far more informative than the graph structure." However, it's not clear what you mean by node attributes are more informative than graph structure. A very clear definition and intuition, examples need to be given."
>
> We respectfully disagree with the reviewer’s statement that “the main conclusion is vague and not so clear”. In the theorems in Sec. 4, we have defined the conditions clearly and rigorously to reach the conclusions we claim.
>
> Specifically, for example, in Theorem 2, we have clearly shown in different regimes defined by comparing the attributed information $\sqrt{m}||\mu - \nu||_2$ and the structural information $|log \frac{p}{q}|$, how large the performance gap between the nonlinear model and the linear model could be.
>
> It is intuitive that $\sqrt{m}||\mu - \nu||_2$ reveals the attributed information because $\mu$ and $\nu$ denote the means of the Gaussian node attributes of two classes respectively. It is also intuitive that $|log \frac{p}{q}|$ reveals the structural information because larger $|log \frac{p}{q}|$ means a larger gap between p and q, which further indicates a larger gap between the probabilities that a node connects to nodes from the same class or the different class.
>
> > "It's not clear what practical implication we can gain? Why not just always add non-linearity, since they will outperform in some cases?"
>
> Our paper is the first paper to systematically analyze the role of non-linearity in GNNs. The significance of our work lies in a more thorough understanding of nonlinearity in GNNs, including its functions, its potential performance gain and a good theoretical explanation of previous empirical observations.
>
> First, we reveal the functions of the nonlinearity in GNNs (1) for node-attribute transformation (2) for propagation from the perspective of statistical analysis. We show that the optimal nonlinearity for node-attribute transformation should depend on the likelihood ratio of node attribute distributions (Eq. (4)). We also show that the nonlinearity for propagation is for information control to avoid the contamination of neighbor’s attributes (Remark 1 and Theorem 2). As a result, the performance gap of nonlinear propagation is significant only when the node attributes are informative, which well explains many previous empirical observations. We argue that all these implications are  significant for researchers and practitioners to understand their GNNs’ behavior in practice.
>
> Regarding how these insights guide a practical model design and why not always use nonlinearity, our results indicate that 1) when one has complicated node attributes (far from Gaussian or NEF distributions), adding non-linear node attribute transformation could be helpful. If node attributes are like Gaussian, using linear node transformation should already give a reasonably good performance; 2) when node attributes are not that informative, one may not expect much performance gain by using nonlinear propagation. Yes, general non-linear functions include linear functions as a special case, so using the former one seems to be always more powerful. However, please remember that using non-linearity will increase the complexity of the model, including the training complexity,  the complexity of infrastructure to serve non-linear models, and the concern on the generalization due to the complex non-linear decision boundary. Because of these concerns, we think when a linear model can give a reasonably good performance, practitioners may want to avoid using non-linear models, and our analysis is significant by shedding some light on the regime where a linear model can give such a good performance compared to the optimal non-linear model.

---

> > ### Author Response · Authors · 2022-08-01
> > **Response to the reviewer's further note on changing the evaluation**
> >
> > We greatly thank the reviewer for letting us know the reason of changing the evaluation and clarifying the caused confusion. We believe it is crucial for all of us to keep the review process clear and transparent. If the reviewer does not have further concerns beyond the originally posted ones, we hope that our response has addressed the reviewer's previous concerns. We are looking forward to the reviewer's reconsideration.

---

### Official Review · Reviewer_6wNk · 2022-07-11

**Rating:** 6
**Confidence:** 3
**Soundness:** 3 good
**Presentation:** 4 excellent
**Contribution:** 4 excellent

**Summary:**

This paper characterizes the effect of non-linearity in GNNs from a Bayesian inference perspective. It shows whether non-linearity is beneficial is related to whether node attributes are more informative than structural information. Experiments are conducted to verify the theoretical results.

**Questions:**

See above.

**Limitations:**

I don’t think this paper has any negative societal impact.

**Strengths And Weaknesses:**

The paper studies a very interesting research question concerning the role of non-linearity in GNNs. It poses valuable theoretical results answering when and why non-linear propagations have the potential to outperform linear counterparts, which is of benefit to the GNN community. The idea of using MAP to study the optimal propagation scheme may be of interest beyond the scope of this paper. The results also seem solid though I didn’t verity proofs in the supplementary material. Extensive experiments are conducted, and the results appear to align with the theoretical findings.

I also think the paper is clearly written and easy-to-follow. I appreciate that the authors tried to give high-level explanations of theories throughout the paper. Yet, there seems to be some typos, e.g., Eq.(3) in line 199 and 203 should be Eq.(4)? I suggest the authors double checking equation references.

However, I have a lingering concern regarding the discrepancy between the models used for analysis (i.e., Eq.(3) and Eq.(7)) and the common ones used in practice (e.g.,  non-linear [1-3], linear [4,5]). Though the models considered in the paper are valid GNN instantiations, the corresponding functions $\phi$ and $\psi_{Gau}$ are different with their counterparts in mainstream GNN models (i.e., $\phi$ is ReLU and $\psi$ is linear map), and usually we also apply activation function to $\psi(X_{v})$ in Eq.(3). It is also strange that baseline linear model in Eq.(7) still uses $\psi$ since it corresponds to using feature transformation in conventional linear GNNs. Whether the theoretical results generalize to other GNN settings is questionable (both theoretically and empirically). Maybe the authors can shed more lights on this?

[1] Semi-supervised classiﬁcation with graph convolutional networks.
[2] Inductive representation learning on large graphs.
[3] Graph attention networks.
[4] Simplifying graph convolutional networks.
[5] LightGCN: Simplifying and Powering Graph Convolution Network for Recommendation.

---

> ### Author Response · Authors · 2022-07-31
> **Response to Reviewer 6wNk**
>
> We greatly thank Reviewer 6wNk for appreciating our contributions to the theory and supporting the acceptance of this work. The reviewer asked very in-depth questions including how our analyzed GNNs align with the practical models.  Here, we respond to the questions  proposed by Reviewer 6wNk.
>
> > "Yet, there seems to be some typos, e.g., Eq.(3) in line 199 and 203 should be Eq.(4)?"
>
> Thank you for pointing out. We have fixed them in the revised version.
>
> > "Though the model considered in this paper is valid GNN instantiations, there is still a gap between the model investigated here and the mainstream GNNs (the corresponding functions $\phi$ and $\psi$ are different with their counterparts in mainstream GNN models). ​​Usually, we also apply activation functions to $\psi(X)$."
>
> We do not think that the difference that the reviewer refers to leads to any changes of our conclusions. First, we may view the activation functions to $\psi(X)$ that the reviewer mentioned as one part of $\psi$ the node-attribute transformation, which is a case covered by our theory in Sec. 3 for the case with general node attributes. In Sec. 3, we show that the optimal node-attribute transformation $\psi$ is nonlinear in general and can naturally include the extra activation function the reviewer mentioned.
>
> In Sec. 4, our analysis focusing linear $\psi$ is because here our main goal is to analyze the nonlinear propagation part $\phi$ and we assume Gaussian node attributes. Once the node attributes are Gaussian, the optimal node-attribute transformation $\psi$ is linear. Adding additional non-linear activation to the node attributes suggested by the reviewer in this Gaussian case will give only suboptimal performance. This will make the comparison between nonlinear and linear propagations less clear, because we do not know whether the performance gap is due to the different propagations or due to the sub-optimal transformation of node attributes.
>
> Moreover, the following works [1,2,3,4], adopt linear propagation. The general GNNs [5,6] adopt non-linear propagation. Comparing the difference in propagations is our main focus in Sec. 4. These models may have non-linear node-attribute transformation or linear node-attribute transformation, comparing which is not our main focus in Sec. 4.
>
> [1] Simplifying graph convolutional networks.
> [2] LightGCN: Simplifying and Powering Graph Convolution Network for Recommendation.
> [3] Predict then Propagate: Graph Neural Networks meet Personalized PageRank
> [4] Adaptive Universal Generalized PageRank Graph Neural Network
>
> [5] Semi-supervised classiﬁcation with graph convolutional networks.
> [6] Inductive representation learning on large graphs.
>
> > "Why the baseline linear model in Eq.(7) still uses $\psi$ since it corresponds to using feature transformation in conventional linear GNNs?"
>
> In Sec. 4, our analysis focuses on analyzing the nonlinear propagation part $\phi$ and we assume Gaussian node attributes. For the Gaussian node attributes, the optimal node-attribute transformation $\psi_{Gau}$ (defined in Eq. (6)) is linear. Therefore, Eq. (7) is a linear model instead of a non-linear model.
>
> In Sec. 3, before we assume Gaussian node attributes or NEF node attributes, the general node-attribute transformation $\psi$ could be nonlinear.

---

> > ### Comment · Reviewer_6wNk · 2022-08-08
> > **Thanks for your response**
> >
> > I thank the authors for the detailed response, and my concerns are addressed. Therefore, I would keep my score which I think is a fair assessment of the paper.

---

### Official Review · Reviewer_48Dg · 2022-07-26

**Rating:** 7
**Confidence:** 3
**Soundness:** 3 good
**Presentation:** 4 excellent
**Contribution:** 3 good

**Summary:**

The paper theoretically and empirically studies non-linearity and linearity in Graph Neural Networks (GNN) from the Bayesian-Inference Perspective and tries to explain the phenomenon that in practice for node classification tasks, the non-linearity of GNNs only has marginal improvement over their linear counterparts. The paper considers CSBM graph and one-hop neighbor settings with proper density and bounded attribution assumptions. The authors claim that the max-a-posterior estimation consists of two types of non-linearity, a non-linear transformation of node attributes and a ReLU-activated feature aggregation from neighbors. The paper also verifies their theorem on synthetic and real-world datasets.

**Questions:**

1. Due to the CSBM setting, the paper can directly get the closed form solution equation (4) and give an analysis. I wonder is there any way to generalize the result, e.g., each node has its private connecting probability.
2. Assumption 1 (Moderate Structural Information) means that the paper only considers a dense regime. Nowadays, community cluster detection works on sparse regime, e.g., S(p,q) = O(1/n) in [1]. Is the analysis still guaranteed in this regime?
3. The non-linearity of a real dataset is hard to measure. Is there any way to metic the non-linearity in the dataset? Figure 4 seems related but still weak.

[1] Li, Xiaodong, Yudong Chen, and Jiaming Xu. "Convex relaxation methods for community detection." Statistical Science 36.1 (2021): 2-15.


**Limitations:**

The paper only considers a 1-hop neighbor. It would be better to consider a 2-hop neighbor. I believe that for a 2-hop neighbor the gap between linearity and non-linearity will be much larger. For the experiments, the scale problem can be solved by sub-sampling.

**Strengths And Weaknesses:**

Strongness:

1. Contribution: As far as I know, this is the first paper systematically study non-linear properties in GNN. The paper provides a clear take-home message that the non-linearity only makes a contribution when the Attributed Information is sufficient (Theorem 2). The theorem is strong enough to support the authors’ claim and seems consistent with our intuition. The observation about the ReLU non-linearity part is insightful. It is closely connected to the empirical GNN. Remark 1 is insightful. The paper also verifies their theorem on synthetic and real-world datasets. They are consistent with the theorem.
2. Writing: The paper has a clear motivation and good structure in Sections 3 and 4. The paper has all the necessary components, e.g., theorem and experiments.

Weakness:

1. The gap in Figure 1 right is not large enough. It would be better to plot with a larger x.
2. The discussion about the non-linearity of node-attribute transformation is limited. It would be better to give one more example for the NN case, e.g., a two-layer network.

Claim: I only checked a limited part of the proof.

---

> ### Author Response · Authors · 2022-07-31
> **Response to Reviewer 48Dg**
>
> We greatly thank Reviewer 48Dg for appreciating our contributions to the theory and supporting the acceptance of this paper. Reviewer 48Dg has also raised several very insightful questions, which we are to respond to next.
>
> > "Due to the CSBM setting, the paper can directly get the closed form solution equation (4) and give an analysis. I wonder is there any way to generalize the result, e.g., each node has its private connecting probability."
>
> Our optimal non-linear model can be generalized to other settings as long as the model satisfies edge-independent assumption (i.e. edges random variables are mutually independent conditioned on the labels of nodes). If this assumption is satisfied, the MAP estimation will give  graph convolution with ReLU activation.
>
> Specifically, Reviewer 48Dg mentions a setting where each node has its own private connecting probability. Based on our understanding, the reviewer refers to the case when $p$ and $q$ vary according to each node. In this case, a solution similar to Equation 4 can be still derived from MAP estimation Equation 3 or Equation 34 (Appendix B), i.e. $\mathcal{P}\_{v} = \psi(X\_{v}; \mathcal{P}\_{1}, \mathcal{P}\_{-1}) + \sum\_{u \in \mathcal{N}\_v}\phi( \psi(X\_{v}; \mathcal{P}\_{1}, \mathcal{P}\_{-1}); \log(p\_{uv} / q\_{uv}))$. In this form, $p$ and $q$ are replaced by $p\_{uv}$ and $q\_{uv}$ to highlight the dependence on the private connecting probability between nodes. So, our analysis is still applied.
>
> However, one should note that in this case, messages from different nodes have edge-dependent parameters ($p\_{uv}$ and $q\_{uv}$) and therefore the MAP algorithm is not permutation invariant. So, the algorithm may not well align well with the commonly-used GNN architecture whose parameters are shared across the nodes.
> > "Assumption 1 (Moderate Structural Information) means that the paper only considers a dense regime. Nowadays, community cluster detection works on sparse regime, e.g., S(p,q) = O(1/n) in [1]. Is the analysis still guaranteed in this regime?"
>
> This is an interesting question. Our current analysis cannot be applied to the sparsest regime, $S(p, q) = \mathcal{O}\_n(1 / n)$ yet. Those community detection works focus on the sparsest regime because their goal is to derive the fundamental limit instead of to understand the behavior of a specific algorithm as ours. To the best of our knowledge, none of the current works (e.g., [1,2]) that analyze GNNs consider a regime sparser than ours. However, we admit that it is interesting to investigate the behaviors of GNNs in that sparse regime in the future.
>
> [1] Baranwal, A., Fountoulakis, K., & Jagannath, A. “Graph convolution for semi-supervised classification: Improved linear separability and out-of-distribution generalization.” ICML 2021.
>
> [2] Lu, Wei. "Learning Guarantees for Graph Convolutional Networks in the Stochastic Block Model." ICLR 2022
>
> > "The non-linearity of a real dataset is hard to measure. Is there any way to metic the non-linearity in the dataset? Figure 4 seems related but still weak."
>
> We guess the reviewer refers to how we can measure the attribute non-linear transformation for a real dataset. Based on our derivation, such a transformation depends on the specific distributions of node attributes in the dataset, specifically, the likelihood ratio of the distributions of node attributes from two classes. Currently, we only understand when those distributions belong to the natural exponential family, such a likelihood ratio is linear, while for the general case, such a ratio is non-linear. However, the performance gap between using linear transformations and non-linear transformations could strongly depend on the specific distributions, where principled knowledge can hardly be derived.
>
> > "The gap in Figure 1 right is not large enough. It would be better to plot with a larger x."
>
> We updated Figure 1 in the paper with larger attributed information ($||\mu - \nu||\_2$). The performance gap between non-linear and linear model is more significant.
>
> > "The discussion about the non-linearity of node-attribute transformation is limited. It would be better to give one more example for the NN case, e.g., a two-layer network."
>
> As we have derived, the optimal node-attribute transformation should depend on the specific distributions of node attributes, which varies so largely across the datasets, and therefore is less principled for analysis than non-linear propagation. We can make the distributions of node attributes arbitrarily complicated to derive arbitrarily complicated nonlinear node-attribute transformations. In the paper, we have used Laplace distribution as an example to show a node-attribute transformation with one non-linear layer. One may build a superimposed Laplace distribution with PDF $p(x) \propto exp( -| ||x- c|| - d| )$ to have a transformation with two non-linear layers, while we do not see much significance by listing these more complicated examples.

---

> > ### Comment · Reviewer_48Dg · 2022-08-07
> > **Response to Authors**
> >
> > I appreciate the authors' response, which answers most of my questions. After checking other reviews' comments, I did not have new concerns. Thus, I tend to keep my original rating.

---

### Author Response · Authors · 2022-07-31
**Response Summary to Reviews**

We appreciate the valuable feedback and insightful comments from all our reviewers. All reviewers agree that we are studying an interesting and important problem. Our results have provided insightful theory paired with solid experiment demonstration, which get particularly appreciated by Reviewer 48Dg and Reviewer 6wNk. There remain some questions on more discussions on non-linear attribute transformations, and on how the model used for analysis in this paper is related to common GNNs. We will address these questions in the following response to each reviewer.

One note is that Reviewer o4Ni initially also gave an acceptance recommendation (with score 5) on the first several days after the review got released. Reviewer o4Ni changed the evaluation to score 3 while not giving further reasons, which caught our attention.

---

### Meta-Review · Area_Chair_H4Mw · 2022-08-26

**Recommendation:** Accept
**Confidence:** Less certain

**Metareview:**

The paper studies the functions of non-linearity in GNNs for node classification tasks, through Bayesian learning. It considers graphs from the statistical model CSBM and shows the max-a-posterior estimation for the label has two types of non-linearity. With Gaussian assumption on the node attributes it further proves that the second type of nonlinearity is only superior when the attributes are far more informative that the graph structure; similarly when there is a distribution shift between training and testing. It also provided verification experiments on synthetic and real data.

The paper considers an important topic and has provided a novel in-depth investigation. Some concerns of the reviewers are:

1. Interpretation of the theoretical results. The theorems are quite technical, and more elaboration on their implications can the readers appreciate better the results. The authors have provided some more clarification in the response.

2. Connections between the theoretical models and the practical ones. The analysis makes some assumptions, e.g., dense CSBM graphs, Gaussian attributes. On the other hand, some simplifying assumptions are necessary for theoretical study. The empirical verification also matches the theory well.

**Award:**

No

---

### Decision · Program_Chairs · 2022-09-14

Accept